# Diagnosis and Management of Rectal Neuroendocrine Tumors (NETs)

**DOI:** 10.3390/diagnostics11050771

**Published:** 2021-04-25

**Authors:** Francesco Maione, Alessia Chini, Marco Milone, Nicola Gennarelli, Michele Manigrasso, Rosa Maione, Gianluca Cassese, Gianluca Pagano, Francesca Paola Tropeano, Gaetano Luglio, Giovanni Domenico De Palma

**Affiliations:** Department of Clinical Medicine and Surgery, University of Naples “Federico II”, 80131 Naples, Italy; milone.marco.md@gmail.com (M.M.); nicogenna@yahoo.it (N.G.); michele.manigrasso89@gmail.com (M.M.); rosamaione95@libero.it (R.M.); gianluca.cassese91@gmail.com (G.C.); gianluca.pagano94@gmail.com (G.P.); fpt.tropeano@gmail.com (F.P.T.); gaetano.luglio@gmail.com (G.L.); giovanni.depalma@unina.it (G.D.D.P.)

**Keywords:** rectal neuroendocrine tumor, endoscopic mucosal resection, endoscopic submucosal dissection, transanal endoscopic microsurgery, endoscopy, surgery

## Abstract

Rectal neuroendocrine tumors (NETs) are rare, with an incidence of 0.17%, but they represent 12% to 27% of all NETs and 20% of gastrointestinal NETs. Although rectal NETs are uncommon tumors, their incidence has increased over the past few years, and this is probably due to the improvement in detection rates made by advanced endoscopic procedures. The biological behavior of rectal NETs may be different: factors predicting the risk of metastases have been identified, such as size and grade of differentiation. The tendency for metastatic diffusion generally depends on the tumor size, muscular and lymphovascular infiltration, and histopathological differentiation. According to the current European Neuroendocrine Tumor Society (ENETS) guidelines, tumors that are smaller than 10 mm and well differentiated are thought to have a low risk of lymphovascular invasion, and they should be completely removed endoscopically. Rectal NETs larger than 20 mm have a higher risk of involvement of muscularis propria and high metastatic risk and are candidates for surgical resection. There is controversy over rectal NETs of intermediate size, 10–19 mm, where the metastatic risk is considered to be 10–15%: assessment of tumors endoscopically and by endoanal ultrasound should guide treatment in these cases towards endoscopic, transanal, or surgical resection.

## 1. Introduction

Rectal neuroendocrine tumors (NETs), formerly known as “carcinoid tumors” because of their peculiar characteristics, are rare, with an incidence of 0.17% during screening colonoscopy [1], but they represent 12–27% of all NETs and 20% of gastrointestinal NETs [2].

The gastrointestinal tract is the most frequent site for the onset of NETs, and the rectum is the second localization by frequency after the small intestine, even if these neoplasms represent only 1–2% of all rectal tumors [3].

Although rectal NETs are uncommon tumors, their incidence has increased over the past few years, as reported in the surveillance, epidemiology, and end results (SEER) database [4], and this is probably due to the improvement made in detection rates by advanced endoscopic procedures and the increased participation in screening colonoscopy programs, rather than a real increase in incidence in the population. Most rectal NETs are asymptomatic and they are found incidentally during colonoscopy performed for colorectal cancer screening, as a small yellowish submucosal lesion with intact overlying mucosa, frequently located in the midrectum (between 4 and 8 cm from the anorectal junction) [5], while approximately 80% of them are 10 mm or less in size and contained within the submucosal layer at diagnosis [6]; however, unlike in the past, they remain neoplasms with a variable potential for malignancy [7].

When they are symptomatic, rectal NETs may occur with diarrhea, abdominal pain, weight loss, or gastrointestinal blood loss [5]; severe anemia, hepatomegaly, or abdominal palpable mass can be associated with metastatic disease [8]. Various clinical data, such as hypertriglyceridemia, low high-density lipoprotein cholesterol levels, higher levels of serum cholesterol, and the presence of metabolic syndrome, have been associated with an increased risk of developing rectal NETs, as reported in the literature; NETs have also been associated with hereditary neuroendocrine syndromes, like von Hippel Lindau syndrome, neurofibromatosis type 1, multiple endocrine neoplasia type 1, and tuberous sclerosis [9].

The biological behavior of rectal NETs may be different: factors predicting the risk of metastases have been identified, such as size and grade of differentiation. The tendency for metastatic diffusion generally depends on the tumor size, muscular and lymphovascular infiltration, and histopathological differentiation [10].

## 2. Diagnosis of Rectal NETs

Most of rectal NETs are diagnosed incidentally during endoscopic evaluation for colorectal cancer screening and they present as small lesions, usually less than 12 mm in diameter, and slightly protruding from the mucosa; these macroscopic features of rectal NETs resemble hyperplastic polyps, making them hard to differentiate from the other most common polypoid lesions. For this reasons, often the diagnosis is known afterwards, because a routine snare polypectomy or mucosectomy is mistakenly performed; therefore, the diagnostic pathway becomes relevant for recognizing these type of lesions and for avoiding mistakes leading to their mismanagement [11].

Frequently, rectal NETs are located in the frontal or lateral wall of the mid-rectum, on average between 4 and 8 cm from the anal verge [12].

Generally, these lesions appear on endoscopic examination as smooth, round, mobile, submucosal nodules, or focal areas of submucosal thickening, covered by a yellow-discolored mucosa (reflecting the presence of chromogranine) [7].

Although this is the most frequent macroscopic manifestation, several studies have shown that rectal NETs can present with various endoscopic features, and the more the appearance of these lesions differs from the canonical one, the higher is the possibility that they are associated with a high risk of metastases. In particular, it is important for the endoscopist to pay attention to subtle surface changes, such as ulceration or erosion, which can be important indicators of aggressive disease [13]. It is necessary to distinguish NETs from other submucosal lesions, like lipoma or myoma, which are usually not treated, and this can be performed using an endocytoscope, an ultra-high magnifying endoscope that allows the visualization of tumor cells in the submucosa, the glandular structure, and cellular atypia in vivo. In endocytoscopy, NETs appear as cells with small round nuclei arranged in a cord-like or honeycomb array [14].

There are contrasting opinions on the management and diagnostic path that follows the incidental feedback of a neuroendocrine neoplasm of the rectum. One school of thought tends to practice biopsies during endoscopic examination, to proceed directly to the histological characterization of the lesion. This approach is not recommended for many reasons: first, a complete resection of neoplasm is only found in about 30% of cases, so a further operational procedure is still necessary [15], then routine biopsies may be ineffective for obtaining tissue for histopathological diagnosis, because rectal NETs are submucosal lesions, and last, random biopsies can produce tissue fibrosis, which may disturb dissection in ESD procedures and suction in EMR-L [16].

According to the current European Neuroendocrine Tumor Society (ENETS) consensus [17], an endoscopic rectal ultrasound (EUS) should be the next and most important diagnostic step after endoscopic exam for a suspected rectal NET.

EUS is a good preoperative exam that defines accurately the tumor size, the depth of invasion, and the presence of pararectal lymph node metastases: this is important for choosing the appropriate therapeutic approach [18]. Moreover, EUS findings, such as lobulated forms, irregular margins, and echogenic foci, may predict a higher grade of malignancy of the submucosal lesions [19].

The rectal wall appears on an EUS as a five-layer structure: the first two, hyper- and hypo-echogenic, respectively, correspond to the mucosa. The third layer is hyperechogenic and corresponds to the submucosa. In EUS, rectal NETs usually appear as hypoechoic lesions, located in both the second and third wall layer, and clearly demarcated from the surrounding tissue [20].

If the rectal NET interrupts the upper two thirds of the third layer of the rectal wall with its hypoechogenic structure, it is classified as SM-D (−); if it extends to the lower third of the submucosa, it is classified as SM-D (+). This classification is useful for both the pre-operative assessment of the lesion and the post-operative assessment of resection margins [21].

EUS is not always able to make a differential diagnosis with other non-carcinoid submucosal lesions and the most similar in appearance is leiomyoma, which also appears as a hypoechogenic submucosal nodule with a uniform internal echo [22]. In these cases, EUS diagnostic accuracy is approximately 50%, and EUS-guided fine needle aspiration (EUS-FNA, Cook Medical Inc., Bloomington, IN, USA) and EUS-guided fine needle biopsy (EUS-FNB, Cook Medical Inc.) can be useful to distinguish between these lesions. EUS-FNA allows obtaining cell blocks for cytological and immunohistochemical typing, with a specificity that grows as the size of lesion increases (>2 cm) [23]. EUS-FNB is a more innovative technique that improves diagnostic accuracy compared to EUS-FNA and consists in making a lesion biopsy to also evaluate a piece of intact architectural structure [24].

At the histological examination, rectal NETs usually show a trabecular growth pattern, with a nest-like or rose-like structure.

The most common immunohistochemical markers for neuroendocrine neoplasms are synaptophisine, chromogranin A, and neuro-specific enolase [25], but the most sensitive marker for rectal NETs is SATB2, which is positive in 88% of rectal NETs, and its positivity confirms a rectal origin in case of metastasis of occult origin [26]. Moreover, insulinoma-associated protein 1 (INSM1) has recently been reported to be an marker specifically expressed by rectal NETs [27].

After EUS evaluation, which allows describing the primary lesion, patients should undergo computed tomography (CT) to discover regional and distant metastases. Magnetic resonance imaging (MRI) can be used when the nature of the injuries found at CT is uncertain. The ENETS consensus recommends CT/MRI when the diameter of the primary lesion is >10 mm or when, after resection, residual and metastatic diseases are suspected.

In high-grade NETs (G3) it is important to perform functional tests, which can be useful for the management of rectal NETs and for distant metastasis detection.

Octreoscan is the first approved radiopharmaceutical for carcinoid tumor imaging; it consists of somatostatin analogs radio-labelled with 111 Indium, used to detect somatostatin receptor positive tissue [17]. Positron emission tomography CT (PET-CT) is another functional imaging, using 68-Ga-labeled octreotide 68-Ga-labeled octreotide; this diagnostic method offers a higher sensitivity than octreoscan for NET detection [28].

However, these functional tests cannot be used for poorly differentiated neoplasms that may not express somatostatin receptors; in these cases, PET-CT with 18 F-fluoro-deoxyglucose (18-FDG) may be more appropriate for metastasis detection [5] (Scheme 1).

## 3. Prognosis of Rectal NETs

Rectal NETs have an excellent prognosis compared to all other neuroendocrine gastrointestinal neoplasms, especially because they are often diagnosed incidentally as small and early neoplasms [5]; even if rectal NETS are mostly low-grading neoplasms and remain asymptomatic for a long time, they still have the possibility of malignant degeneration, so it is necessary to evaluate the lymphovascular invasion, defined as the presence of tumor cells in the blood or lymphatic vessels, for all neuroendocrine neoplasms of the rectum [29].

Data from the SEER database show that the 5-year survival rate is 88.3 %, depending on tumor size and invasion [4], and according to the ENETS consensus and the American Joint Committee on Cancer (AJCC), rectal NETs need to be classified by grading and by staging with TNM [17,30].

The grading is based on mitotic index per 10 high-power fields (HPFs) and the expression of Ki67, a tumor proliferation marker:-Low Grade (G1): 2 mitotes/10 HPFs and <2% Ki67 index;-Intermediate Grade (G2): 2–20 mitotes/10 HPFs and 3–20% Ki67 index;-High grade (G3): 20 mitotes/10 HPFs and >20% Ki67 index.

The staging with TNM includes tumor size, depth/invasion of tumor, and metastatic spread [5] (Table 1 and Table 2).

As reported in the literature, the major criteria for assessing the invasiveness of rectal NETs are the size of the tumor, the depth of invasion, increased mitotic index, and lymphovascular invasion, but the only one that can be evaluated pre-operatively and that can guide the treatment approach is the size of the lesion [31].

Rectal NETs less than 10 mm in size at diagnosis have a very low risk of distance metastasis (<3%), and local excision can be curative in most cases; long-term outcomes are excellent, with a 5-year survival of 98–100%, while rectal NETs diagnosed >20 mm and those with regional (N1) and distant metastases (M1) have a worse prognosis, with survivals of 54–74% and 15–37%, respectively [5].

## 4. Management of Rectal NETs

The choice of therapeutic intervention for rectal NETs depends on their features, especially on their size, grade of differentiation, muscolaris propria involvement, lymphatic and vascular invasion, increased tumor proliferative index, and risk of metastasis.

The aim of the treatment is to achieve a complete oncological resection, with clear margins and no residual disease, therefore endoscopic resection is indicated if there is no evidence of invasion beyond the submucosa and presence of regional disease. In case of evidence of invasion involving the muscolaris propria and regional disease, surgery must be considered as the first option [5].

According to the current European Neuroendocrine Tumor Society (ENETS) guidelines, tumors that are smaller than 10 mm and well differentiated are thought to have a low risk of lymphovascular and muscolaris invasion and rarely exhibit malignant potential, and should be completely removed endoscopically; the risk of metastases has been estimated at less than 3% for rectal NETs less than 10 mm in size. Rectal NETs larger than 20 mm are likely to have a higher risk of involvement of the muscularis propria and high metastatic risk (60–80%) and are candidates for surgical resection. There is controversy over rectal NETs of intermediate size 10–19 mm, where the metastatic risk is considered to be 10–15%: assessment of tumors endoscopically and by endoanal ultrasound should guide in these cases towards endoscopic, transanal, or surgical treatment [17,32]. 

Various endoscopic techniques can be used for treatment of rectal NETs, including endoscopic polypectomy, endoscopic mucosal resection (EMR), modified EMR (m-EMR), and endoscopic submucosal dissection (ESD).

m-EMR includes: endoscopic mucosal resection with a ligation device (EMR-L), which in some studies is also called endoscopic submucosal resection with band ligation (ESMR-L); cap-assisted EMR (EMR-C); EMR after circumferential pre-cutting (EMR-P), which in some studies is also called endoscopic mucosal resection with circumferential incision (CIEMR) or circumferential submucosal incision prior to endoscopic mucosal dissection (CSI-EMR).

The choice of optimal endoscopic treatment depends on the tumor characteristics, such as size, and mucosal and submucosal appearance [33].

Endoscopic polypectomy is not usually used in rectal NETs resections, as it often does not provide adequate and complete resection of the lesion, and additional interventions may be needed [34]. 

EMR is an endoscopic technique based on a submucosal injection of a saline solution to elevate the mucosal lesion away from the muscularis propria, followed by a snare cautery resection. EMR has been frequently used in the resection of small and superficial neoplasms confined to the mucosa or superficial submucosa because of its simplicity and lesser invasiveness, but its feasibility is still uncertain in cases of rectal NETs because of the potential problem of incomplete excision [33].

Several studies report that complete resection rates of standard EMR are variable, ranging from 30% to 70% [35], and recent meta-analyses demonstrated that m-EMR and ESD are superior in terms of complete resection rate, defined as the resection of the entire tumor in one piece (also called en bloc resection), and histologic complete resection rate, defined as the en bloc resection with tumor-free margins and no lateral or vertical margin involvement of the resected specimen, compared to conventional EMR [33,36].

EMR can also be performed using a dual-channel endoscope (EMR-D) and, by lifting the lesion with grasping forceps to identify the lower margin of the submucosal lesion and strangling the base with a snare, EMR-D enables deeper resection compared with standard EMR. Lee et al. [37] compared the efficacy of EMR-D with ESD, showing that EMR-D is a safe and effective technique for resection of rectal NETs < 16 mm in size, with complete and histological resection rates similar to ESD, but with a procedure time significantly shorter.

m-EMR techniques are characterized by the use of special devices, which allow for a better resection of the tumor; EMR-C and EMR-L are commonly performed, and while EMR-P was recently introduced, several studies [38,39,40,41] have shown that it is a safe and effective technique for resection of rectal NET, with successful en bloc resection and histologic resection rates, and it may be preferable to conventional EMR [39]; while compared to ESD, it has similar outcomes in terms of achieving complete resection, but with a shorter procedure time and lower rate of complications [40].

EMR-P is performed by lifting the mucosa with a saline injection, making a circumferential incision (pre-cutting) using the tip of the snare or special endoknives and resecting the tumor with a snare; this technique presents advantages over other m-EMR procedures because it has no size limitation with respect to the tumor resection [38].

EMR-L has some advantages over EMR, particularly due to the use of a ligation device; it is performed after an initial submucosal saline injection to elevate it from the muscle layer, by suctioning the lesion into the ligating device and cutting around the concerned area with the ligation by using a round snare [42]; while in the EMR-C technique, endoscopic suctioning of the tumor is performed with a transparent cap fitted to the scope, followed by closure of a snare looped along the inner ridge of the cap [35].

EMR-C is an effective technique for endoscopic resection of rectal NETs, as demonstrated in several studies [43,44] and may be preferable for rectal NET resection because it is technically easier and faster than ESD. However, Lee et al. compared the two techniques, EMR-L and EMR-C, concluding that EMR-L may be the preferable treatment method, considering both endoscopic en bloc resection rate and histologic complete resection rate [35].

Mashimo et al. [45], in a retrospective analysis of a small group of patients, demonstrated that ESMR-L is an effective and safe endoscopic technique for the resection of rectal NETs, and it is better than conventional EMR or endoscopic polypectomy for complete resection, especially in the lower rectum, where the rectal wall is thick and surrounded by connective tissue, and ESMR-L can be performed with full suction to achieve a deeper vertical margin, without complications.

Kim et al. [10] compared the same two techniques in a larger group of patients, showing that ESMR-L is significantly superior to EMR in terms of the complete resection rate of small rectal NETs, regardless of the tumor location, in upper or lower rectum.

A disadvantage of ESMR-L may be the fact that it is only applicable for tumors of 10 mm or less in size, due to the short diameter of the caps fitted to colonoscopies [10].

In agreement with the superiority of ESMR-L over EMR, in terms of achieving complete resection, other studies have focused on the comparison between ESMR-L and ESD.

ESD is an advanced endoscopic technique used for en bloc resection of slightly invasive gastrointestinal tumors or larger mucosal lesions, and it provides a precise histological staging. ESD is performed by submucosal dissection after a submucosal injection and a circumferential cutting of the mucosa around the lesion. It has been reported that ESD is an effective and safe technique for treating gastrointestinal tumors, including rectal NETs, even if it is associated with a high risk of complications (perforation and bleeding), a long procedure time, and required technical skills compared to other endoscopic resection techniques [46].

Several studies [34,37,47,48,49] and metanalyses [33,36] have reported that ESD is a superior modality to EMR for the en bloc and histologically complete resection of rectal NETs, while there is no significant difference between ESD and modified EMR techniques, in terms of achieving complete resection, even if it is associated with increased procedural time and rate of complications compared to EMR, and requires a high level of endoscopic experience [34].

Another study, conducted by Niimi et al. [42], compared ESD with EMR-L, showing that EMR-L is an effective endoscopic technique and has a similar rate of complete resection in cases of small rectal NETs compared to ESD, but with a short time of procedure and hospitalization, a lower rate of complication, and not requiring additional technical skills, as in ESD.

Bang et al. [6] even demonstrated in their analysis the superiority of EMR-L over the ESD technique for the treatment of small rectal NETs (<10 mm), with an optimal en bloc resection rate and a significantly higher pathological complete resection rate in comparison to ESD (100% vs 54.2%), but this was probably also due to inappropriate specimen preparation, as the authors reported in the paper.

Wang et al. [1] proposed a hybrid ESD as a safe and effective treatment for rectal NETs as an alternative to conventional ESD. Hybrid ESD is a simplified technique where, after the circumferential incision around the lesion, the submucosal injection of a saline solution, and a partial dissection of the submucosa, snaring is performed using a polypectomy snare to dissect the lesion completely, instead of an endoknife. In their study, they showed that hybrid ESD achieved high rates of en bloc resection and histologic complete resection, without increases in adverse events and with a shorter procedure time compared to ESD (Table 3).

Yan et al. [50] in their study compared the short-term clinical outcomes between ESD and transanal local excision (TALE), showing that pathologically complete resection rates were optimal for both techniques (97% for ESD and 100% for TALE), but the complete resection of rectal NETs can be achieved when using a proper procedure according to the tumor size; TALE may be more appropriate in cases of local recurrence, even if it is more invasive and associated with higher morbidity.

TALE is a surgical technique performed in T1 malignant rectal tumors, including rectal NETs, located in the lower rectum, less than 7 cm from the anal verge and less than 1/3 lumen diameter in size; it is not used in the higher rectum because the exposure is significantly limited. TALE is performed by introducing anal retractors into the anal canal to maintain exposure, lifting the mucosa with a saline injection to elevate the lesion and resecting the tumor with electrocautery under direct vision; the defect in the rectal wall is closed with an absorbable suture [50].

An alternative to TALE is transanal endoscopic microsurgery (TEM), a minimally invasive technique that allows the complete resection of benign or T1 malignant rectal neoplasms, including rectal NETs, with a safety surgical margin and a continuous layer; TEM is also a feasible surgical option indicated for complete removal of residual tumor, in case of tumor-positive resection margins or recurrences [51].

TEM is performed by using a multi-channel port positioned transanally that allows, at the same time, the use of a rigid rectoscope with magnified three-dimensional vision and endosurgical instruments: the scheduled resection area is previously marked by electrocautery dots, then a full-thickness resection down to the perirectal fat is performed, and finally the defect in the rectal wall is closed by a continuous running suture with clips or absorbable monofilament. As with TALE, TEM must be performed with the patient under local or general anesthesia. TEM shows several advantages over both endoscopic technique, by providing a full-thickness resection of large tumors, and over conventional transanal resection, because it provides improved operative visualization and access to lesions higher in the rectum [52].

The morbidity of TEM reported in literature ranges from 4% to 29%, and the most common complications are bleeding, peritoneal entry with peritonitis, rectal wall perforation, fecal incontinence, and conversion to laparotomy [53].

Moore et al. [54] compared the effectiveness of TEM and TALE for rectal neoplasms, showing that TEM is the first choice for local excision of rectal tumors and it is more likely to yield clear resection margins and lower recurrence rates, while TALE may be the first therapeutic choice for scar embedded rectal NETs.

McCarty et al. [55] in their meta-analysis compared ESD with TEM, reporting that the two procedures are similar in terms of resection rate, complications, and recurrence in case of large rectal tumors, even if ESD is associated with a significantly shorter procedure time and duration of hospitalization

EMR and ESD achieved a complete microscopic resection in 46.3% to 65.5% and in 75% to 82.6% of cases, respectively. TEM allows achieving a 100% rate of free resection margins [56].

Another endoscopic technique introduced recently is endoscopic full thickness resection (EFTR), which is performed in lesions that are difficult to resect, both for anatomic location and for negative lift sign after submucosal injection. EFTR is performed using a full thickness resection device (FTRD), that is an over-the-scope system, which allows a single step EFTR after placement of a modified over-the-scope-clip (OTSC). Limitations correlated with this technique are the resectable tumor size, due to diameter of the FTRD (<20 mm, but this is not a strong limitation because rectal NETs >20 mm are referred to surgery), but EFTR appears to be an effective and safe resection method for well-differentiated rectal NETs of smaller size; combining a high complete resection rate with low complication rate and short procedure time [57].

In addition, Meier et al. [58] demonstrated in their study the high complete resection rate of EFTR, even if the technique was reported to be more difficult and associated with lower full-thickness resection rates in the rectum compared with the colon.

Surgery is recommended in case of tumors >20 mm or between 10 mm and 19 mm with high risk features, such as muscular or lymphovascular invasion, or if the margins are positive on endoscopic, or for local resection. Laparoscopic surgery has become the standard surgical approach because it provides better short- and long-term outcomes compared with open surgery and, often, low anterior resection or intersphincteric resection, depending on the localization of the tumor, with total mesorectal excision being performed. Many rectal NETs arise near the anus, and colectomy with lymphadenectomy often necessitates rectal transection with a colostomy, although anal preservation is an important outcome of radical surgery in terms of quality of life [52,59].

The prognosis after endoscopic resection of rectal NETs is generally good, but, although a macroscopically complete resection is achieved in most cases, microscopically remnant NETs may be present on resection margins, and this may be the cause of local tumor recurrence or distant metastasis. According to the ENETS guidelines, lesions presenting a histologic non-complete resection after endoscopic treatment must undergo additional salvage treatments, but this does not always happen, as the positive resection margins are not always predictive factors of local recurrence or metastasis. Sung et al. [60] analyzed the success rate of endoscopic complete resection and the long-term prognosis of endoscopically resected rectal NETs, and they showed that endoscopic resection is an optimal treatment for rectal NETs smaller than 15 mm, confined to the submucosal layer, and it may be curative in cases with complete histological resection, with an excellent long-term prognosis and 100% 5-year survival, while in cases in which complete histological resection cannot be achieved, additional endoscopic treatment or surgical resection can be used supportively and can lead to an excellent outcome.

Cha et al. [61] evaluated the prognosis of endoscopically resected rectal NETs after non-complete resection and clarified factors determining the additional salvage treatment, such as positive resection margins and lymphovascular invasion. They showed that endoscopically resected rectal NETs with a non-curative resection had a good prognosis, regardless of additional salvage treatments, while patients with lymphovascular invasion needed to undergo radical surgery with lymph node dissection. The risk factors for lymph node metastasis are tumor size >14 mm, increased mitotic rate, and lymphovascular invasion; also small rectal NETs can have lymphovascular invasion and subsequent lymph node metastasis, but they tend to have excellent short-term prognosis, with only an 0.3% recurrence rate during the 5-year follow-up period, and they do not require surgical intervention, but only a long-term follow up of 10 to 20 years to assess for any delayed recurrence, as demonstrated in the meta-analysis conducted by Kang et al. [62].

In addition Moon et al. [63] in their study confirmed that not all patients with non-curative resection underwent additional salvage treatments, because they did not modify prognosis and long-term outcomes.

According to ENETS guidelines [32], regular follow-up is not required for rectal NETs completely resected and less than 10 mm in size with low metastatic risk, while rectal NETs completely resected and >10 mm should undergo surveillance rectoscopy after 1 year, 3 years, and then every 5 years. In case of factors predictive of local or distant recurrence, a rectoscopy or EUS examination should be performed every 6 months over 3 years and annually thereafter (Scheme 2).

## 5. Conclusions

Although rectal NETSs are still uncommon tumors, their incidence has increased over the past few years, due both to the widespread use of colonoscopy for screening and due to improvements in detection rates made by advanced endoscopy. The choice of therapeutic intervention for rectal NETs depends on their features, especially on their size, grade of differentiation, muscolaris propria involvement, lymphatic and vascular invasion, increased tumor proliferative index, and risk of metastasis; and the assessment of tumors endoscopically and by endoanal ultrasound should guide treatment approach.

Rectal NETs smaller than 10 mm in size are usually low grade and can be successfully treated endoscopically or by local excision, because they rarely metastasize and represent 80% of all rectal NETs diagnosed.

EMR-L should be considered as the first-line of treatment for small rectal NETs, because it is a safe and effective technique, relatively simple, and less time-consuming compared with ESD. ESD should be left as a second-line therapy when EMR-L is not applicable. Rectal NETs that are 10–20 mm in diameter are associated with a poorer prognosis compared with those <10 mm, because they have a higher risk of muscolaris and lymphovascular invasion, and they can be treated by endoscopic or transanal resection or radical surgery. TEM is more likely to yield clear resection margins and lower recurrence rates compared to TALE.

Rectal NETs > 20 mm in size have a high metastatic potential and they are candidates for radical rectal resection and lymph node dissection. The duration of follow-up remains uncertain, as data on long term outcomes of rectal NETs are scarce.

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
