# Peer review of "Diagnosis and Management of Rectal Neuroendocrine Tumors (NETs)"

_diagnostics, 2021, doi:10.3390/diagnostics11050771_

Round 1

Reviewer 1 Report

This was a comprehensive review of rectal NETs which include the prognosis, diagnosis and management.

Did the authors find that their review of the literature supported the ENETS current guidelines? Are there limitations regarding data available to support current guidelines? The review seemed to address all the various sources of data but then the abstract summarized current ENETS guidelines and not the author's practice.

There was mention of the significance of presentation with ulceration. Was this characteristic mentioned in any of the other endoscopic reviews. 

Line 50, the use of the word anamnestic is incorrect

Line 151, the use of word diagnosticated should be changed to diagnosed. 

Line 417, there is a random 'e' in the middle of the sentence.

Author Response

Review Report 1: In this manuscript, all references support ENETS guidelines and these guidelines are constantly updated, also considering the continous increase in incidence of rectal neuroendocrine tumors. Our clinical practice is not mentioned in the abstract as it is a literature review.
Regarding the presentation with ulceration of rectal NETs, this characteristic is not mentioned in other endoscopic reviews, but I find this peculiarity in the article: " Kim BN, Sohn D. Atypical endoscopic features can be associated with metastasis in rectal carcinoid tumors. Surg Endosc 2008, 22, 1992-6", as mentioned in the manuscript at reference 13.
I change the word "anamnestic" to "clinical" at line 50 , the word "diagnosticated" to "diagnosed" at line 151 (that in the new revised manuscript became line 155) and the word "e" to "and" at line 417 (that in the new revised manuscript became line 427)

Reviewer 2 Report

This article is well written including recently published data. Table 1 and Scheme 2 are very good.  However, the confusion of the usage of NETs and carcinoid tumor is seen in this article. According to the recently published data (in Table 1), NETs are divided into G1 (same as carcinoid tumor)  G2 (not carcinoid tumor), and G3 (not carcinoid tumor). So, in this article, in the Introduction Section, above mentioned recent division of NETs shall be described (NET G1 = carcinoid tumor), and thereafter, NET G1 (not usage of carcinoid tumor)  shall be used. This changing of the terminology lead to well recognition of the readers.

Author Response

 Review Report 2: In the manuscript, I use the term "neuroendocrine tumor (NET)" as synonymous with "carcinoid", as indicated in the Introduction section. Grading G1, G2, G3 refers to the differentiation grade of the tumor and it is not related to the name. In the article, I use both "NET" and "carcinoid" to avoid multiple repetitions. However, I have modified the manuscript using only the term NET to avoid confusion for the readers.

Regarding the changes made in the manuscript, I report the line number and exact changes below:
Line 48, 69, 96, 111, 153, 262, 269, 284, 343, 370, 421 change "carcinoid" to "NET"
Line 50 change "anamnestic" to "clinical"
Line 155 change "diagnosticated" to "diagnosed"
Line 427 change "e" to "and"